# COMPLEXITY-LIMITED MULTI-TASK TRAINING FOR COMPOSITIONAL EMERGENT COMMUNICATION

## ABSTRACT

Human languages are largely compositional: sentences derive meaning based on the meanings of constituent words. Conversely, emergent communication systems, learned by unsupervised neural networks, rarely learn human-like compositionality. To encourage compositionality, we propose a new training method that combines information-bottleneck losses with a multi-task framework. By training on a diversity of tasks, we induce task-specific vocabulary; by penalizing complexity, we decrease redundancy and entanglement in communication. Our information-theoretic framing explains results from studies in noisy-channel emergent communication, and outperforms recent population-based training methods. Our work thus address important theoretical questions in compositional communication and achieves state-of-the-art results.

## 1 INTRODUCTION

We seek to train agents to learn compositional emergent communication (EC) that decomposes according to human-like factors. In traditional EC literature, cooperative agents are trained in partially observable environments, with the ability to communicate (Lowe et al., 2017; Mordatch & Abbeel, 2018). By training to maximize a task-specific reward or utility function, communication "emerges."

Unfortunately, unlike typical human communication, emergent communication is rarely compositional. For example, humans might refer to colorful shapes by decomposing messages into one word for color and one for shape (e.g., "red square"). Conversely, agents often learn to combine factors into a single symbol (e.g., one symbol for red triangles, another symbol for red squares, etc.). Furthermore, even if agents do communicate meanings via a combination of symbols, rather than learning one symbol for "red" that can combine with any shape, agents often learn to interpret that same symbol differently in different messages (e.g., as "red" for squares and as "blue" for triangles). While significant prior literature has considered the problem of inducing human-like compositional communication (Kottur et al., 2017; Kuciński et al., 2021; Chaabouni et al., 2020), recent research tends to focus on generalization or compositionality, without comparison to human meanings (Chaabouni et al., 2021b; Rita et al., 2022).

In this work, we propose a computational model for the emergence of compositional communication based on a combination of pragmatics and communicative efficiency. Pragmatics considers the role of context or goals in communication and, in EC, often gives rise to communication that solves a specific task well but is non-compositional (Goodman & Frank, 2016). We propose that training EC agents on a *distribution* of tasks, as opposed to a single task, can partially address such non-compositionality. To further encourage compositionality, we incorporate a cognitively-inspired communicative efficiency term to penalize the complexity of communication; this biases agents to avoid redundant communication (Gibson et al., 2019; Zaslavsky et al., 2018; Tucker et al., 2022). Overall, therefore, we model how compositional communication may arise due to multi-task training (inducing a vocabulary that is suitable for many tasks) and pressures for communication efficiency (encouraging re-use of vocabulary in multiple contexts).

We make two primary contributions in this work. First, we extend an existing metric to better align with our judgement of compositionality. Second, we propose a complexity-limited multi-task training framework to improve compositionality in emergent communication, which we show outperforms methods from prior art.

## 2 RELATED WORK

In an effort to induce more "human-like" emergent communication (EC), several researchers have sought to induce compositionality in emergent communication. Kottur et al. (2017) found that restricting the vocabulary size of communicative neural agents was a necessary but not sufficient change for inducing compositionality. Overly-large vocabularies allowed agents to combine multiple factors into a single vocabulary element (e.g., communicating about red squares via one symbol, and red triangles via a different symbol). Unfortunately, small vocabularies alone are insufficient for inducing compositional communication according to human-like factors, as agents may learn to decompose meanings along a variety of orthogonal bases, which may not align with human understanding (Locatello et al., 2019).

Numerous EC works have encountered this issue of alignment of compositionality with human-specified factors. Some works induce generalizable EC and that achieves high compositionality metrics, but such compositionality does not appear to align with human-defined factors (Chaabouni et al., 2020; 2021b; Karten et al., 2023). To induce more *human-like* compositionality, Rita et al. (2022) train populations of agents and find that larger, and more heterogeneous, populations of agents lead to more compositional communication (as measured by a metric of topographic similarity, which we explore later). At the same time, while considering just a single pair of agents, Kuciński et al. (2021) show that adding a small amount of noise to the communication channel improves compositionality. Lastly, Gupta et al. (2020) examined effects of varying speaker capacity (e.g., the vocabulary size) and found no benefit to limiting capacity for inducing compositional communication.

In our work, we seek to induce human-like compositionality in emergent communication, via a combination of information bottleneck methods and a multi-task training framework. Our information bottleneck approach provides a principled way to limit the information conveyed at each timestep, supplanting hardcoded methods that artificially limit agent vocabulary or add noise (Kuciński et al., 2021; Kottur et al., 2017), and unlike Gupta et al. (2020) we *do* find benefits to limiting complexity of communication. At the same time, our work complements population-based training methods.

## 3 BACKGROUND

In this paper, we use existing compositionality metrics and evaluate neural architectures that support complexity-bounded communication. Here, we review the technical details of relevant prior work.

### 3.1 COMPOSITIONALITY METRICS

We consider two metrics of compositionality that have been widely used in prior literature: `topsim` and `posdis`. `topsim` measures the "topographic similarity" between inputs and communication by measuring if similar inputs give rise to similar communication (Brighton & Kirby, 2006; Lazaridou et al., 2018). More formally, it is the Spearman correlation coefficient between the Levenshtein distance for each pair of messages and the Euclidean distance for each pair of inputs. A `topsim` value of 0 indicates no correlation; a positive value, up to maximum of 1.0, indicates that similar messages represent similar inputs.

`posdis` is the "positional disentanglement" of communication and measures the specificity of information about a particular field in the input (e.g., color) with a particular timestep in communication (e.g., second symbol in a message) (Chaabouni et al., 2020). It is computed as

$$\texttt{posdis} = \frac{1}{L} \sum_{i \in [1,L]} \frac{I(s_i; f_1^i) - I(s_i; f_2^i)}{H(s_i)} \tag{1}$$

where $L$ is the length of the message, $s_i$ refers to the symbol communicated at timestep $i$, $f_i^1$ is the feature with the highest mutual information with $s_i$ ($f_i^1 = \arg\max_{f \in \mathcal{F}} I(s_i; f)$), and $f_i^2$ is the feature with the second-highest mutual information with $s_i$. `posdis` is bounded between 0 and 1, with a value of 1 indicating that each position is informative of only one feature in the input.

### 3.2 COMPLEXITY-BOUNDED EMERGENT COMMUNICATION ARCHITECTURES

In our experiments, we use the Vector-Quantized Variational Information Bottleneck – Categorical (VQ-VIB$_\mathcal{C}$) method for generating complexity-limited discrete tokens (Peng et al., 2023). A VQ-VIB$_\mathcal{C}$ speaker agent is parametrized via a feedforward encoder, $h$, and a set of discrete tokens, $\zeta$. Given an input, $x$, the speaker generates a continuous latent representation, $z = h(x) \in \mathcal{R}^Z$, which, for a message of length $L$, is divided into $L$ representations: $z_i \in \mathcal{R}^{Z/L}$. Lastly, a symbol at timestep $i$ is stochastically selected with probability $\mathbb{P}(s_i = \zeta_j | x) \propto \exp ||z_i(x) - \zeta_j||^2$. These discrete representations are concatenated into the overall message.

Via its stochastic discretization process, VQ-VIB$_\mathcal{C}$ allows for variational bounds on the complexity of communication, but prior work has either considered single-timestep communication (Tucker et al., 2022) or used multiple discrete representations in non-communication settings (Peng et al., 2023). In our work, we apply VQ-VIB$_\mathcal{C}$ and other neural architectures in multi-timestep EC settings for the first time.

## 4 TECHNICAL APPROACH

In this work, we make two primary contributions: 1) we generalize the existing `posdis` metric to better align with intuitions of compositionality, and 2) we propose a complexity-limited multi-task training framework to increase compositionality.

### 4.1 GENERALIZATION OF POSITIONAL DISENTANGLEMENT

Rather than use the standard `posdis` metric, we propose a modified measure of compositionality, positional disentanglement - mutual information (`pdmi`), defined as

$$\texttt{pdmi} = \frac{1}{L} \sum_{i \in [1,L]} \frac{\max_{f \in \mathcal{F}} I(s_i; f)}{\sum_{f \in \mathcal{F}} I(s_i; f)} \tag{2}$$

where $I(s_i; f)$ represents the mutual information of feature $f$ with the symbol emitted at timestep $i$. The `pdmi` metric is closely related to `posdis` as it depends upon the most informative feature for a given timestep, and is maximized at 1.0. There are two important differences, however.

First, rather than compute the difference between the top-two most informative symbols, we only use the most informative feature. This mitigates unintuitive behaviors that the `posdis` metric can exhibit. For example, consider two speaker agents that communicate about three features: A, B, and C. If one speaker communicates about A and B at $t = 1$, but not C, whereas the other speaker communicates about all three fields at $t = 1$, they would have the same `posdis` value despite the second speaker's communication being more entangled. Conversely, as desired, `pdmi` is higher for the first speaker, reflecting more compositional communication.

Second, we set the denominator in `pdmi` to be the sum of the mutual information about all fields at a given timestep. When `posdis` is normalized by the entropy over symbols, this artificially lowers the disentanglement metric, even when information about distinct features is communicated at precise timesteps in a message. Trivially, if two symbols encode the same information (much like synonyms in natural language), this increases $H(s)$; we believe that the presence of such synonym-like communication should not be measured as lower compositionality. Conklin & Smith (2023) also note that variation in ways of communicating may lower `posdis` measures even for highly compositional communication. At the same time, some work suggests that EC agents naturally minimize the entropy of communication (Kharitonov et al., 2020); if so, the denominators in `pdmi` and `posdis` are equivalent (although we did not find that to be the case in our experiments). We include several simple examples illustrating the distinction between `posdis` and `pdmi` in Appendix A.

### 4.2 COMPLEXITY-LIMITED MULTI-TASK TRAINING

We proposed a complexity-limited multi-task training framework for training agents to learn compositional communication, depicted in Figure 1. Our framework comprises three parts, each of which alone is not sufficient for compositional communication:

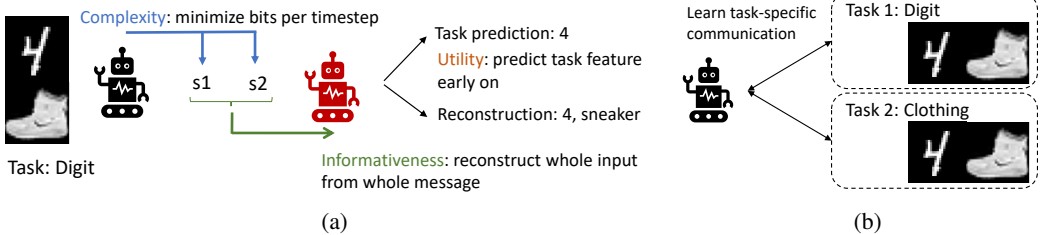

(a)                                                                    (b)

Figure 1: Our proposed complexity-limited multi-task training framework. a) For a given task, a speaker observes a multi-feature input and a task id, and communicates via multiple symbols ($s1$, $s2$, etc.) to a listener. A complexity loss penalizes the number of bits transmitted, a utility loss encourages early prediction of the task-specific feature, and a reconstruction loss encourages the overall reconstruction of the input, given the full message. b) By training on a distribution of tasks, we induce task-specific vocabulary, which is combined into compositional communication.

- **Complexity regularization**: limiting communication complexity decreases redundancy.
- **Multiple tasks**: training "impatient" agents according to a distribution of human-specified tasks induces communication about specific fields.
- **Informativeness**: an informativeness loss causes agents to learn accurate reconstructions of all fields via a combination of symbols.

Each term affords intuitive interpretations, which we corroborate in experiments. Without limits on complexity, there is no pressure for agents to learn re-usable components that may be combined into overall meanings. Thus, penalizing complexity biases agents towards learning compositional communication, although not necessarily along desired axes. To induce agents to communicate according to human-specified fields, we use a multi-task framework, where each task consists of agents predicting the value of a specific field. By training on multiple tasks, with impatient agents (where "impatient," as introduced by Rita et al. (2020), signifies a pressure to make a correct prediction in few timesteps), we force agents to learn field-specific vocabulary, for multiple fields separately. Lastly, we train agents via an informativeness loss to predict all fields, based on the (unordered) complete message; this encourages agents to combine field-specific symbols into an overall message. The three terms – limiting complexity, learning field-specific communication, and enabling overall understanding of a message – also align with pressures thought to guide natural languages: communicative efficiency, pragmatics, and semantics, respectively (Gibson et al., 2019; Zaslavsky et al., 2018; Goodman & Frank, 2016; Zaslavsky et al., 2020).

Our overall training loss, combining these three terms, is included in Equation 3.

$$\max \quad \mathbb{E}_{f \in \mathcal{F}} \left[ \lambda_U \sum_{t \in [1,L]} U(Y_f, \hat{Y}_f(S_{1:t})) - \lambda_C \sum_{t \in [1,L]} I(X; S_t) + \lambda_I I(X; \hat{Y}(S_{1:L})) \right] \quad (3)$$

Given inputs, $X$, a speaker outputs communication, $S$, for a message over $L$ timesteps. At each timestep, a listener makes a prediction about field $f$ (specified by the task) based on all previous communication, $\hat{Y}_f(S_{1:t})$, which is compared to the desired prediction $Y$, according to a utility function $U$. Also over each timestep, the complexity of communication, measured as the mutual information between inputs and communication at that timestep, $I(X; S_t)$, is penalized. We maximize the informativeness of communication, measured as the mutual information between the input and the listener's prediction of all of the input's features, based on the full communication vector. Scalar weights regulate the relative importance of the utility, complexity, and informativeness terms. Lastly, we instantiate our multi-task framework by taking the loss in expectation over fields to predict.

In practice, training to directly optimize Equation 3 is challenging due to the complexity term. Therefore, as in prior literature, we use neural speaker architectures that support variational bounds on complexity ($I(X; S_t)$) (Tucker et al., 2022). The listener is implemented via a transformer-based architecture, allowing us to remove positional encoding information from the message (see

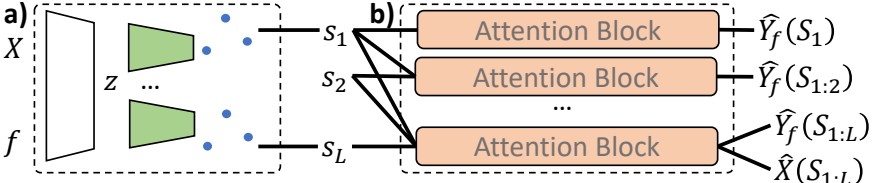

Figure 2: Speaker (a) and listener (b) architectures used in experiments. The speaker generates a latent representation, $z$, from input $X$ and field id $f$. $z$ is divided into $L$ parts and passed through identical communication heads (in green) to generate communication symbols $s_1...s_L$. The listener takes in such symbols, attending to prior timesteps, and makes field predictions at each timestep ($\hat{Y}_f$) as well as reconstructs the overall input based on the full message ($\hat{X}$).

high-level architectures in Figure 2, and implementation details in Appendix D). Lastly, while one can use a variety of utility functions, $U$, in our experiments, we use a supervised loss to predict categorical fields, $Y_f$.

### 4.3 THEOREM ON REPETITION

Given the training framework proposed in the previous section, here, we show how a combination of explicit pressures and inductive biases can give rise to compositional communication. First, Theorem 1 states how our multi-task framework, in combination with positive $\lambda_C$, biases agents towards learning to communicate about fields in a single timestep:

**Theorem 1 (Penalizing complexity prevents repetition)** *Within our training framework, if $\lambda_U > \lambda_C$, and $\lambda_C > 0$, agents will communicate about the task-specific field only at $t = 1$.*

We include a proof in Appendix B. Briefly, for a sufficiently large $\lambda_U$ to outweigh the complexity regularization term, agents will communicate about the task-specific field at $t = 1$; for $t > 1$, there is no additional benefit to communicating redundant information about that field and, assuming $\lambda_C > 0$ and independent features, doing so incurs a cost.

Theorem 1 states that agents will learn a vocabulary that supports communicating complete information about each field, $f$, in a single timestep. This is an important, but insufficient, step towards compositional communication, as agents could communicate about multiple fields in the same timestep. One could explore further losses or architectural choices, such as limiting the vocabulary size, but such changes still do not guarantee desired compositionality (Locatello et al., 2019). Thus, we rely upon the evidence from our experiments to validate whether our complexity-limited multi-task framework, in combination with the inductive biases of the neural architectures we use, are enough to induce compositional communication.

## 5 EXPERIMENTS

We performed experiments in three domains, extending environments from prior literature. In all experiments, speakers observed an input, $x$, as well as a feature id, $f$; the listener predicted the value of the feature specified by $f$, and reconstructed $x$. In training, we fixed $\lambda_I = 1.0$ and, by varying $\lambda_C$ and $\lambda_U$, found that our complexity-limited multi-task framework led to the greatest compositionality.

In all experiments, we trained feed-forward agents by backpropagating the utility, reconstruction, and complexity losses. Prior literature has established that REINFORCE-based training and backpropagation converge to similar results, although backpropagation appears to converge faster and more stably (Chaabouni et al., 2021a; 2020; Tucker et al., 2022). We tested speaker architectures based on Gumbel-Softmax (GS) (Jang et al., 2017; Maddison et al., 2017), VQ-VIB$_{\mathcal{N}}$, and VQ-VIB$_{\mathcal{C}}$ (Tucker et al., 2022; Peng et al., 2023) (see Appendix C for details of architectures). In the main paper, we present results using VQ-VIB$_{\mathcal{C}}$ as it performed best and focus on compositionality metrics. Results for GS and VQ-VIB$_{\mathcal{N}}$ models, as well as additional metrics, including reconstruction accuracy, are included in Appendix E.

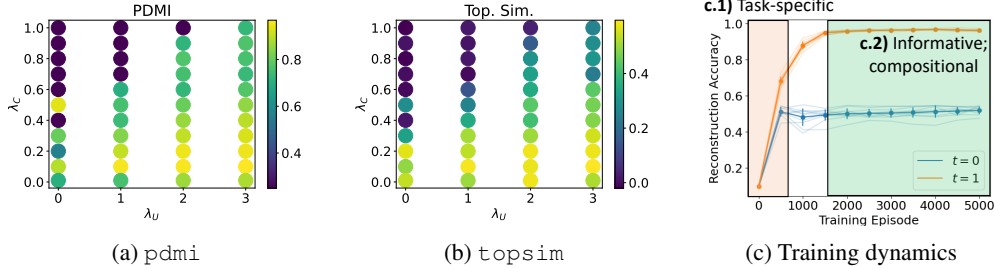

(a) `pdmi`  (b) `topsim`  (c) Training dynamics

Figure 3: Median `pdmi` (a) and `topsim` (b) results, over 5 trials, for varying utility and informativeness pressures in the Symbolic 2D domain. Small increases in $\lambda_C$ catalyzed greater compositionality compared to traditional training ($\lambda_U = \lambda_C = 0$). c) In training, agents first learned task-specific communication (c.1) before learning to compose multiple fields over time (c.2).

## 5.1 SYMBOLIC - 2D

First, we discuss results from our simplest symbolic domain, inspired by Rita et al. (2022)'s experiments, included to provide intuition for later findings. Inputs comprised two symbolic fields, each taking on one of ten possible values; agents had two timesteps to communicate, using a vocabulary of size 100. Borrowing notation from Rita et al. (2022), we re-write this scenario as $\mathcal{K} = 2, \mathcal{V} = 10, \mathcal{W} = 100, L = 2$. Intuitively, this scenario corresponds to our motivating example of communicating about colors and shapes. Implementation details for training, in this and other domains, are included in Appendix D.

Results from our experiments, for different combinations of $\lambda_C$ and $\lambda_U$ are included in Figure 3 a and b. In each figure, the location of a point represents the $\lambda_C$ and $\lambda_U$ values used in training; the color of each point represents the value of the reported metric.

**Comparing compositionality metrics** Figures 3 a and b show how small increases in $\lambda_C$ increased compositionality and that `pdmi` is a more sensitive measure of compositionality than `topsim`. Notably, setting $\lambda_C = \lambda_U = 0$ (corresponding to no complexity or field-specific pressures) led to `pdmi` of $0.57$ $(0.06)$ (medians and standard errors over 5 trials reported). Conversely, using our proposed complexity-limited multi-task training via $\lambda_C = 0.1, \lambda_U = 3.0$, `pdmi` reached a value of nearly $1.0$, implying near-perfect compositionality. At the same time, even for the high-`pdmi` agents with $\lambda_C = 0.1, \lambda_U = 3.0$, `topsim` was only $0.57$, far below the maximum theoretical value of $1.0$. Median `posdis` was also only $0.53$ $(0.01)$ (Figure 9 in Appendix E). The differences between compositionality metrics show important benefits of using `pdmi`: for $\lambda_C = 0.1, \lambda_U = 3.0$ agents clearly communicated about just one field at each timestep (per the `pdmi` metric); thus, the low values of `topsim` and `posdis` reveal limitations of such metrics in reflecting compositionality.

**Non-monotonic benefits of $\lambda_C$** Figures 3 a and b also confirm an important bound on useful values for $\lambda_C$. As $\lambda_C$ increased past 0.1, both `pdmi` and `topsim` decreased, to a minimum value at $\lambda_C = 1.0$. Recall that models were trained with $\lambda_I = 1.0$; it is therefore unsurprising that models learned uninformative (and non-compositional) communication for $\lambda_C = 1.0$. By sweeping over $\lambda_C \in [0, 1]$, we covered the full range of meaningful communication (confirmed by reconstruction accuracy metrics, included in Appendix E). When conducting a Mixed Linear Effects Model (MLEM) test estimating the effect of increasing $\lambda_C$ for small ($\leq 0.1$) and large ($> 0.1$) values, increasing $\lambda_C$ a small amount significantly increased `pdmi` ($p = 0.002$), but increasing $\lambda_C$ past 0.1 decreased `pdmi` ($p < 0.001$). Further details of the statistical test, for this and other domains, are included in Appendix F. Overall, this non-monotonic benefit of penalizing complexity corroborates trends established by Kuciński et al. (2021), who found that adding a small amount of noise to communication increased compositionality. Such a noise-based approach may be thought of as an indirect way of imposing an information bottleneck, which we do directly.

**Training dynamics** Figure 3 c depicts learning dynamics that resulted in compositional communication (generated here for $\lambda_C = 0.1, \ \lambda_U = 3.0$). Each curve represents the decoder's accuracy at

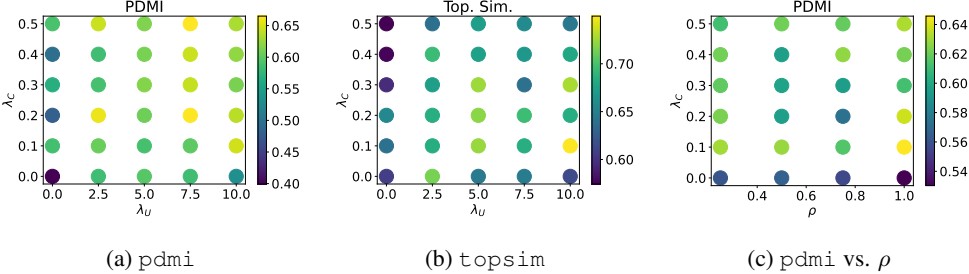

|                  |                  |                        |
| :--------------: | :--------------: | :--------------------: |
| (a) pdmi         | (b) topsim       | (c) pdmi vs. $\rho$    |

Figure 4: pdmi (a) and topsim (b) in the Symbolic 4D domain, for varying $\lambda_U$ and $\lambda_C$. Compared to standard approaches ($\lambda_C = \lambda_U = 0$) compositionality increased for positive complexity ($\lambda_C$) and multi-task training ($\lambda_U$) weights. (c) Varying the listener's learning rate ($\rho$) had minimal effects on compositionality, whereas increasing $\lambda_C$ consistently helped (results for $\lambda_U = 10.0$).

different timesteps, as a function of training episode. That is, the blue line represents the accuracy of the decoder's output after receiving one symbol, and the orange line represents the accuracy after receiving two tokens (the full message). Early in training (highlighted as phase c.1), agents learned field-specific communication and achieved approximately 50% reconstruction accuracy (because the decoder could only identify one of the two fields correctly). In phase c.2, agents moved beyond field-specific communication to improve reconstruction accuracy at the second timestep. Notably, reconstruction accuracy after one timestep remained at 50%, indicating that agents still only communicated about one field at a time, and composed meanings about multiple fields across time. This corroborates the high pdmi value found for this training setup.

**Population training effects**   We implemented a population-heterogeneity baseline based on Rita et al. (2022)'s approach by training teams of size $N$, with learning ratio $\rho$ specifying the likelihood of a gradient update for the listener agent at a given training step.[1]  Rita et al. (2022) found that increasing $N$ and decreasing $\rho$ led to greater topsim values. We refer to the original paper for further details; in general, this population-based approach is complementary to our own.

Overall, we largely reproduced Rita et al. (2022)'s results but found that our approach induced greater compositionality. In the base case $N = 1$, $\rho = 1.0$, Rita et al. (2022) found that agents learned communication with topsim $= 0.51$ (in their paper, see Figure 13, Appendix F). This nearly exactly matches our results without any complexity or multi-task pressures: for $\lambda_C = \lambda_U = 0$: topsim $= 0.53$ (0.02), pdmi $= 0.71$ (0.06). Keeping $\lambda_C = \lambda_U = 0$, for $N \in [1, 2, 4, 8]$; $\rho \in [0.25, 0.5, 0.75, 1.0]$, we found the greatest topsim value of 0.60 (0.05) (for $N = 2$, $\rho = 0.25$) and the greatest pdmi value of 0.89 (0.07) (for $N = 1$, $\rho = 0.25$). The topsim value is almost exactly equal to the maximum topsim value found by Rita et al. (2022), who never surpassed 0.59 for any population size. However, by setting $\lambda_C = 0.1$; $\lambda_U = 3.0$, our agents achieved pdmi $= 0.96$ (0.02) and topsim $= 0.65$ (0.02), outperforming all tested $N$ and $\rho$. Unfortunately, when training agents via a combination of population dynamics (varying $N$, $\rho$ as before) and varying $\lambda_C$, $\lambda_U$, we found no benefit beyond simply varying $\lambda_C$, $\lambda_U$ alone. That is, we always observed maximum compositionality for $N = 1$, $\rho = 1.0$. Given the apparent complementary nature of our approaches, the negative result of combining our methods warrants further exploration.

## 5.2   Symbolic - 4D

We conducted similar experiments in a larger symbolic domain, also from Rita et al. (2022), with $\mathcal{K} = 4$, $\mathcal{V} = 4$, $\mathcal{W} = 100$, and $L = 4$. That is, there were 4 fields, taking on one of 4 values, a vocabulary of size 100, and 4 timesteps. Results from such experiments corroborated the trends established in the Symbolic 2D and are included in Figure 4.

As before, a small but positive $\lambda_C$, in combination with our multi-task framework, resulted in the greatest compositionality. Traditional methods, with $\lambda_C = \lambda_U = 0$, achieved median pdmi $= 0.40$ (0.01) and topsim $= 0.61$ (0.02) (Figures 4 a and b). Conversely, for $\lambda_C = 0.1$; $\lambda_U = 10.0$,

---

[1]Rita et al. (2022) define $\rho_S$ as the likelihood ratio of the speaker updating; we define $\rho$ as the inverse.

Table 1: Population heterogeneity effects for [4, 4, 4, 4] domain (medians and standard errors over 10 trials). Regardless of population dynamics ($N$ and $\rho$), we achieve the greatest compositionality metrics by increasing $\lambda_C$ to limit communication complexity. All results for $\lambda_U = 10.0$.

| $N$ | $\rho$ | $\lambda_C$ | Recons. | `topsim` | `pdmi` | 90% Recons. Eps. |
|---|---|---|---|---|---|---|
| 1 | 1 | 0 | 1.00 (0.00) | 0.61 (0.02) | 0.53 (0.01) | 1000 (63) |
|  |  | 0.1 | 0.99 (0.01) | **0.75 (0.01)** | **0.65 (0.02)** | 1000 (126) |
|  | 0.5 | 0 | 1.00 (0.00) | 0.66 (0.02) | 0.57 (0.01) | 1000 (106)) |
|  |  | 0.1 | 0.99 (0.00) | 0.70 (0.02) | 0.63 (0.02) | 1500 (162) |
| 4 | 1 | 0 | 1.00 (0.00) | 0.61 (0.02) | 0.54 (0.01) | 7000 (369) |
|  |  | 0.1 | 0.98 (0.00) | 0.72 (0.01) | 0.58 (0.03) | 6500 (387) |
|  | 0.5 | 0 | 1.00 (0.00) | 0.66 (0.02) | 0.56 (0.01) | 6000 (293) |
|  |  | 0.1 | 0.98 (0.00) | 0.72 (0.01) | 0.62 (0.01) | 8500 (308) |

`pdmi` = 0.62 (0.03) and `topsim` = 0.76 (0.01). Mixed effects modeling, testing the effects of $\lambda_C$ on `pdmi`, confirmed that increasing $\lambda_C$ significantly increased `pdmi` ($p < 0.001$). We note that reconstruction accuracy decreased as $\lambda_C$ increased (see Appendix E), indicating an important tradeoff between inducing compositionality while retaining information.

As before, we evaluated Rita et al. (2022)'s population training method as a complementary baseline and found that our framework alone afforded the greatest benefits. Results from such experiments are included in Figure 4 c and Table 1. First, Figure 4 c shows how decreasing $\rho$, the ratio of gradient updates for the listener agent vs. the speaker agent, for $N = 1$ had little to no effect on `pdmi`, while simply setting $\lambda_C = 0.1$ continued to consistently increase compositionality.

Results from Table 1 corroborate such trends for different values of $N, \rho$, and $\lambda_C$. Full results for $N \in [1, 2, 4, 8]$ are included in Appendix E. Across population sizes and learning rates, penalizing complexity increased compositionality: for every $N$, `topsim` and `pdmi` values were maximized for $\lambda_C = 0.1$. For $\lambda_C = 0$, we reproduced some of the trends established by Rita et al. (2022): decreasing $\rho$ increased both `topsim` and `pdmi`. However, as in the Symbolic 2D domain, we found that increasing $\lambda_C$ had a greater effect, which masked the more subtle population-based effects. Lastly, as highlighted in the rightmost column of Table 1, population-based training with greater $N$ took longer to converge, indicating a greater computational cost for such methods.

While results from this larger symbolic domain largely corroborate earlier trends, it exposes some limitations of our current approach. Agents never achieved perfect `pdmi` or `topsim` scores in this domain. Such sub-optimal performance appears to be the result of optimization failures. In particular, upon inspecting the communication of trained agents, we found that speakers tended to favor a particular communication order (e.g., always $f_1$ at timestep 1, $f_2$ at timestep 2, etc...) even as the training task changed, which, per Theorem 1, should change the order of communication. This behavior emerged early in training and was therefore likely due to initialization effects. Fixed-order communication aligns somewhat with natural languages (e.g., English speakers typically prefer to say "the big red ball" instead of "the red big ball") but worsens `pdmi`. We look forward to investigating this phenomenon in future work.

### 5.3 MNIST

In our final experiments, we tested agents in an image domain and, once again, found that the combination of multi-task training and complexity penalization led to compositional communication. The speaker agent observed two images stitched together, comprising one image from the FashionMNIST dataset (e.g., sneakers) and one image from the MNIST digit dataset (e.g., a handwritten "1"). Agents had a vocabulary of size 100 and $L = 2$. We sought to induce compositional communication in which agents communicated separately about items of clothing and digits.

Figure 5 a shows patterns from trained speakers' compositional communication. The leftmost column designates the training task, specifying whether the utility function was based on predicting the digit or clothing item class. For a given task, and across timesteps in the message, we measured the mutual information between the speaker's communication and the image classes. For example, for the "Digit" task, symbols communicated at $t = 1$ shared 2.9 bits about the digit in the image (top row) but only 0.1 bits about the item of clothing (second row). In the second timestep (rightmost

| Training task | Feature | $t = 1$ | $t = 2$ |
|---|---|---|---|
| Digit | Digit | 2.9 (0.0) | 0.2 (0.0) |
| | Fashion | 0.1 (0.0) | 2.3 (0.0) |
| Fashion | Digit | 0.1 (0.0) | 2.6 (0.0) |
| | Fashion | 2.6 (0.0) | 0.1 (0.0) |

(a) Mutual information (in bits) between features and symbols across timesteps. Done for $\lambda_C = 0.1$; $\lambda_U = 3.0$, pdmi = 0.96.

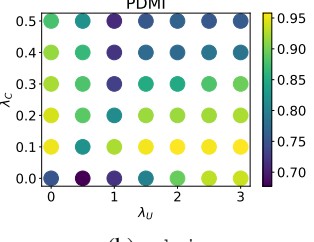

(b) pdmi

Figure 5: MNIST results. a) Measuring the mutual information between features and symbols over time, there is positional disentanglement and re-ordered communication depending upon the training $f$ (medians and std. err. for $\lambda_C = 0.1, \lambda_U = 3.0$). b) pdmi for varying $\lambda_C, \lambda_U$. Results mirror trends from the symbolic domains: small but positive $\lambda_C$ increased compositionality.

column), communication switched to be highly informative about clothing (2.3 bits) but not about digit (0.2 bits). Lastly, the speaker's communication order depended upon training task: for the "Fashion" task, agents communicated first about clothing at $t = 1$ and then about digit. Overall, these results, generated for $\lambda_C = 0.1, \lambda_U = 3.0$, show how agents learned to separate information about different features into different timesteps.

Beyond the specific example considered in Figure 5 a, Figure 5 b depicts pdmi for varying $\lambda_C$ and $\lambda_U$. Results confirmed that our multi-task framework, with complexity penalization, induced desired disentangled communication. For example, a traditional EC setup ($\lambda_C = \lambda_U = 0$) resulted in median pdmi= 0.75 (0.06), but controlling complexity and using the multi-task framework via $\lambda_C = 0.1, \lambda_U = 3.0$ led to pdmi = 0.96 (0.00). That is, having a small penalty on complexity, and a loss encouraging task-specific communication, led to near perfect compositionality. Once again, mixed linear effects statistical testing confirmed that increasing $\lambda_C$ from 0 to 0.1 significantly increased pdmi ($p < 0.001$). Lastly, we conducted population-based experiments, fixing $\lambda_C = 0.1; \lambda_U = 3.0$ and sweeping over $N \in [1, 2, 4, 8]$; $\rho \in [0.25, 0.5, 1.0]$. We found no benefit to using larger $N$ or lower $\rho$: peak pdmi occurred for $N = 1$; $\rho = 1.0$. Overall, results from this domain corroborate trends from our symbolic domains, while further demonstrating that agents can learn compositional communication from pixel-based inputs.

## 6 CONTRIBUTIONS

We proposed a complexity-limited multi-task framework for inducing compositional emergent communication. By training on a distribution of tasks, agents learned field-specific symbols; by minimizing complexity, agents learned to combine field-specific symbols into overall messages. Our work complements concurrent techniques in compositionality, while resolving outstanding questions on why complexity limits alone do not lead to desired behaviors. Lastly, our approach may be interpreted as a cognitive model of how pragmatic communication, in combination with pressures for communicative efficiency, gives rise to compositional human languages.

We look forward to future work building upon our step towards compositional communication. In our work, we make several assumptions about training tasks, such as access to the full distribution of tasks during training. Relaxing this assumption, perhaps to only include information about some fields, could provide important insight into partial alignment of emergent communication and language. In addition, some of our results raise interesting questions about combining multiple EC training frameworks: our approach is theoretically complementary to population-based methods, but in our experiments we did not find benefits to combining multiple approaches.

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
