## A    EXAMPLES OF COMPOSITIONALITY METRICS

In our main paper, we proposed "positional disentanglement - mutual information" (pdmi) and claimed that better corresponds with notions of compositionality than posdis, introduced by Chaabouni et al. (2020). Here, we include simple examples of communication systems, each of which is at least somewhat compositional, to compare posdis and pdmi measures for the same communication systems.

**Example 1: distinctions among partial compositionality**    Agents are trained in a symbolic domain with 4 binary fields (A, B, C, D) and communicate over 4 timesteps (similar to our Symbolic 4D experiments). Table 2 depicts examples of three possible communication schemes in this domain.

| $t =$ | 1 | 2 | 3 | 4 |
|---|---|---|---|---|
| A | 1 | 0 | 0 | 0 |
| B | 0 | 1 | 0 | 0 |
| C | 0 | 0 | 1 | 0 |
| D | 0 | 0 | 0 | 1 |

(a) posdis = 1.0;
    pdmi = 1.0

| $t =$ | 1 | 2 | 3 | 4 |
|---|---|---|---|---|
| A | 1 | 0 | 0 | 0 |
| B | 1 | 0 | 0 | 0 |
| C | 0 | 0 | 1 | 0 |
| D | 0 | 0 | 1 | 0 |

(b) posdis = 0.0;
    pdmi = 0.375

| $t =$ | 1 | 2 | 3 | 4 |
|---|---|---|---|---|
| A | 1 | 0 | 1 | 0 |
| B | 1 | 0 | 1 | 0 |
| C | 1 | 0 | 1 | 0 |
| D | 1 | 0 | 1 | 0 |

(c) posdis = 0.0;
    pdmi = 0.25

Table 2: The number of bits about a field (rows) at a timestep (columns) for different communication protocols. Communication in (a) is perfectly compositional, as reflected by both metrics. Communication in (c) is less compositional than in (b) (agents communicate about all features at timesteps 1 and 3), but only pdmi decreases while posdis remains constant.

Within each table, each entry in row $f$ and column $i$ represents $I(s_i, f)$: the number of bits about a feature, $f$, at timestep $i$ in a message. For example, in Table 2 a, the speaker communicates one bit about field A at timestep 1, one bit about field B at timestep 2, etc.. This type of compositional communication leads to high values for both posdis and pdmi.

More subtle effects arise, however, when communication is not perfectly disentangled. In Table 2 b, the speaker communicates one bit about fields A and B at timestep 1, and one bit about C and D at timestep 3. Such communication is clearly non-compositional in some ways, given that communication about multiple fields occurs at the same timestep. At the same time, the communication depicted in Table 2 c is clearly still less compositional, with communication about all four fields occurring at timesteps 1 and 3. Despite this important behavioral difference between the two communication schemes, posdis remains constant in both cases at 0.0. Recall that posdis only considers the two most informative fields at any given timestep, and therefore ignores any further entanglement. Conversely, pdmi does capture meaningful differences between b and c. Thus, pdmi appears more sensitive to variations in compositionality by not relying only upon two fields.

**Example 2: synonyms and entropy**    Agents are trained to communicate about two binary fields, A and B, with a vocabulary of size eight: $s_i \ \forall \ i \in [1, 8]$. The speaker always communicates about A at timestep 1 and about B at timestep 2; this corresponds to perfectly compositional communication. We consider two variants of this sort of speaker, focusing on communication at the first timestep, depicted in Table 3.

Each entry Table 3 represents the speaker's probability of outputting a particular symbol (designated by column) for a given input (designated by row) at the first timestep in the message. In Figure 3 a, the speaker follows a deterministic policy: outputting $s_1$ if $A = 0$, and outputting $s_3$ if $A = 1$. As a result, both posdis and pdmi $= 1.0$, showing perfect compositionality.

However, in Figure 3 b, we see the limitations of the posdis metric. The speaker in this example is stochastic: choosing $s_1$ and $s_2$ with equal probability if $A = 0$, and choosing $s_3$ and $s_4$ with equal probability if $A = 1$. This stochastic behavior is similar to randomly choosing among a set of synonyms, which does not seem to detract from the compositionality of natural language. However, because such stochasticity increases the entropy of the distribution over vocabulary elements, posdis decreases for Table 3 b. Conversely, as desired, pdmi remains constant. Thus, whereas

| $[A, B]$ | $\mathbb{P}(s_1)$ | $\mathbb{P}(s_2)$ | $\mathbb{P}(s_3)$ | $\mathbb{P}(s_4)$ | $[A, B]$ | $\mathbb{P}(s_1)$ | $\mathbb{P}(s_2)$ | $\mathbb{P}(s_3)$ | $\mathbb{P}(s_4)$ |
|---|---|---|---|---|---|---|---|---|---|
| $[0, 0]$ | 1.0 | 0 | 0 | 0 | $[0, 0]$ | 0.5 | 0.5 | 0 | 0 |
| $[1, 0]$ | 0 | 0 | 1.0 | 0 | $[1, 0]$ | 0 | 0 | 0.5 | 0.5 |
| $[0, 1]$ | 1.0 | 0 | 0 | 0 | $[0, 1]$ | 0.5 | 0.5 | 0 | 0 |
| $[1, 1]$ | 0 | 0 | 1.0 | 0 | $[1, 1]$ | 0 | 0 | 0.5 | 0.5 |

(a) `posdis` $= 1.0$; `pdmi` $= 1.0$       (b) `posdis` $= 0.5$; `pdmi` $= 1.0$

Table 3: A speaker's probability distribution of emitting a symbol, $s_i$, at timestep 1 depending upon different values of $[A, B]$. In both (a) and (b), communication is perfectly disentangled: the speaker only communicates about the value of $A$ at $t = 1$. However, because the entropy of the speaker's distribution is greater in (b) than in (a), `posdis` decreases.

in the previous example `pdmi` was more sensitive to desired changes when aspects of compositionality changed, here we showed that `pdmi` remains invariant to some unimportant changes in communication.

Overall, as illustrated in these two simple examples, we believe that `pdmi` better aligns with human notions of compositionality and therefore should be used instead of `posdis` in future research.

## B  PROOF OF THEOREM 1

In the main paper, we related complexity to redundancy of communication in Theorem1 1; here, we include a proof.

**Proof of Theorem 1** *Within our training framework, if $\lambda_U > \lambda_C$, and $\lambda_C > 0$, agents will communicate about the task-specific field only at the first timestep.*

Our proof follows by 1) writing our training objective for a feature, $f$, 2) decomposing terms within the objective according to features and symbols emitted across timesteps, and 3) regrouping terms in a given timestep, establishing relevant pressures for communication at the first timestep and later timesteps.

$$\textit{maximize} \quad \lambda_U \sum_{t \in [1,L]} U(Y_f; \hat{Y}_f(S_{1:t})) - \lambda_C \sum_{t \in [1,L]} I(X; S_t) + \lambda_I I(X; \hat{Y}(S_{1:L}))$$

$$\textit{maximize} \quad \lambda_U \sum_{t \in [1,L]} I(Y_f; \hat{Y}_f(S_{1:t})) - \lambda_C \sum_{t \in [1,T]} I(X; S_t) + \lambda_I \sum_{i \in \mathcal{F}} I(X; \hat{Y}_i(S_{1:L}))$$

$$\textit{maximize} \quad \lambda_U \sum_{t \in [1,L]} I(Y_f; \hat{Y}_f(S_{1:t})) - \lambda_C \sum_{i \in \mathcal{F}} \sum_{t \in [1,L]} I(Y_i; S_t) + \lambda_I \sum_{i \in \mathcal{F}} I(X; \hat{Y}_i(S_{1:L}))$$

$$\textit{maximize} \quad \lambda_U I(Y_f; \hat{Y}_f(S_1)) - \lambda_C \sum_{i \in \mathcal{F}} \sum_{t \in [1,L]} I(Y_i; S_t) + \lambda_I \sum_{i \in \mathcal{F}} I(X; \hat{Y}_i(S_{1:L}))$$

*For $t = 1$*

$$\textit{maximize} \quad (\lambda_U - \lambda_C) I(Y_f; S_1)$$

*For $t > 1$.  Now assume $I(Y_f; S_1) = H(Y_f)$*

$$\textit{maximize} \quad -\lambda_C \sum_{i \in \mathcal{F}} \sum_{t \in [1,L]} I(Y_i; S_t) + \lambda_I \sum_{i \in \mathcal{F}} I(X; \hat{Y}_i(S_{1:L}))$$

$$\textit{maximize} \quad -\lambda_C \sum_{i \in \mathcal{F}} \sum_{t \in [2,L]} I(Y_i; S_t) + \lambda_I \sum_{i \in \mathcal{F}/f} I(X; \hat{Y}_i(S_{1:L}))$$

The first line restates the training objective for a particular feature, $f$. In the second and third lines, we decompose complexity and informativeness into sums over features. We note that the third line assumes that features are statistically independent, conditioned on $X$; this is true in the domains we consider in our experiments (e.g., digit and clothing) but may not be true in general. In the fourth line, assuming a sufficiently-large vocabulary, we replace the utility sum with a single term, reflecting the fact that the speaker will communicate about $f$ at the first timestep.

We then consider communication at just the first timestep. The informativeness loss is ignored, as it is computed based on the full communication, $S_{1:L}$. Therefore, we find that agents will communicate about feature $f$ at timestep 1 if $\lambda_U > \lambda_C$. Intuitively, this states that, if utility pressures are greater than penalties on complexity, the models will communicate about $f$.

Lastly, we consider communication about feature $f$ in later timesteps. Having already established that agents will communicate about $f$ at $t = 1$, the utility term is already maximized for all timesteps $t > 1$ (as the listener always has access to earlier tokens). We therefore ignore the utility term in the maximization and trade off complexity and informativeness pressures. Here, too, we leverage the fact that $s_1$ contains complete information about $f$, so there is no increase in informativeness by communicating about $f$ in later timesteps. Thus, there are no positive pressures to communicate about $f$ in later timesteps, but there is a complexity-penalization term (assuming $\lambda_C > 0$), so agents will not communicate about $f$ for $t \geq 2$.

Overall, we have shown that, assuming $\lambda_U > \lambda_C$ and $\lambda_C > 0$, agents 1) will communicate about feature $f$ at $t = 1$ and 2) will not communicate about $f$ for $t \geq 2$.

## C  Alternative Architectures

In the main paper, we presented results using the VQ-VIB$_\mathcal{C}$ speaker architecture, but our complexity-limited multi-task framework may be applied to different speakers supporting variational bounds on complexity. In this section, we discuss the VQ-VIB$_\mathcal{N}$ and GS speaker architectures that we tested as additional baselines. Results from such architectures are included in Appendix E and largely corroborate the trends we observed for VQ-VIB$_\mathcal{C}$, although different inductive biases associated with the different architectures had some effect on results.

### C.1  VQ-VIB$_\mathcal{N}$

In addition to VQ-VIB$_\mathcal{C}$, Peng et al. (2023) proposed the Vector-Quantized Variational Information Bottleneck – Normal (VQ-VIB$_\mathcal{N}$) method, named after the fact that it samples from a Normal distribution. We adapted the VQ-VIB$_\mathcal{N}$ method to EC settings.

Similar to VQ-VIB$_\mathcal{C}$ agents, a VQ-VIB$_\mathcal{N}$ speaker is parametrized by a feedforward encoder, $h$, and a set of discrete tokens, $\zeta$. Given input $x$, the speaker generates a continuous representation, $h$. $h$ is mapped via separate linear layers to parameters of a Normal distribution, $\mu(x), \Sigma(x) \in \mathcal{R}^Z$, as in standard Variational Auto-Encoder (VAE) architectures (Kingma & Welling, 2013). A continuous latent representation is sampled from a Normal distribution, using the reparametrization trick: $z \sim N(\mu(x), \Sigma(x)) \in \mathcal{R}^Z$ . Lastly, $z$ is divided into $L$ representations of equal size and discretized by selecting the closest element of the learnable codebook, $\zeta$. By penalizing the KL divergence between the normal distribution and a prior (in our cases, as is standard, fixed to a unit Normal), we penalize the complexity of communication (Higgins et al., 2016). For further details, we refer to Peng et al. (2023) and Tucker et al. (2022), who describe the sampling and discretization processes, although in a distinct setting than emergent communication domains.

### C.2  Gumbel Softmax

We extended the traditional Gumbel-Softmax (GS) architecture to output $L$ onehot vectors, similar to the concurrent quantized vectors output by VQ-VIB$_\mathcal{C}$ and VQ-VIB$_\mathcal{N}$ methods. Concretely, we used the same encoder architecture as for the VQ-VIB models to compute a continuous latent representation, $z$. This representation is split into $L$ evenly-sized parts of size $V$, and each part is passed through a gumbel-softmax layer to generate a $L, V-$dimensional vectors (Maddison et al., 2017; Jang et al., 2017). The gumbel-softmax layer uses a straight-through estimator to allow backpropagation through discrete one-hot tokens, and is commonly used in EC domains (Chaabouni et al., 2020; Rita et al., 2022; Kuciński et al., 2021).

# D  IMPLEMENTATION

In the main paper, we omitted details about the exact implementations we used in experiments; here, we share further information about neural architecture parametrizations and hyperparameters. Anonymized code for all experiments is available here.

## D.1  SPEAKER

We used two types of speaker encoders in our experiments, depending upon the type of input, in combination with the three types of speaker head architectures (VQ-VIB$_\mathcal{C}$, VQ-VIB$_\mathcal{N}$, and GS).

Across domains, we fed the feature id, $f$, into a linear layer of size 32. In parallel, we fed the input, $x$, (e.g., the two MNIST images stitched together) into a feedforward neural network. In the symbolic domains, this feedforward network was a single layer of dimension 32. In the MNIST domain, this network comprised three fully-connected layers of dimensions 256, 128, and 32, with ReLU activations between layers. The outputs of the feature id embedder and the input embedder were concatenated and fed a two-layer neural network (with ReLU activation in between) to generate a continuous latent representation, $z$.

The continuous representation, $z$, was transformed into the EC message, $S$, according to the speaker architecture we were testing. For VQ-VIB$_\mathcal{C}$, for example, we used a VQ-VIB$_\mathcal{C}$ speaker head to split $z$ into $L$ continuous representations and discretize them (likewise for VQ-VIB$_\mathcal{N}$ and GS). The VQ-VIB architectures used codebook sizes dictated by each domain (in all cases, $V = 100$), and we set the dimensionality of $\zeta$ to 10. For GS speakers, $V$ inherently specifies the dimensionality of communication.

## D.2  LISTENER

We used the same listener implementation for all speaker architectures and all domains (subject to changes in dimensions to match the communication and prediction dimensions). The core part of the listener architecture was a three-layer transformer encoder, using four attention heads and hidden dimension 32. We did not use positional encoding to avoid undesired variation in interpretation of the same symbol in different positions in a message. The feature predictor head was instantiated as a single-layer neural network, mapping from the output of the transformer encoder to the appropriate dimension (e.g., 10-dimensional for predicting one of the features in the Symbolic 2D domain). The decoder head was composed of $|\mathcal{F}|$ single-layer neural networks, similarly mapping to the desired dimensionality (e.g., for the Symbolic 2D domain, two prediction heads, each of dimension 10).

## D.3  TRAINING PARAMETERS

In all experiments, we used an Adam optimizer with learning rate 0.001 and otherwise default parameters. (Although note that in our population-based experiments, as $\rho$ varied, we varied how often we updated the listener agent's parameters.) Similarly, all experiments used the same batch size of 1024.

In the Symbolic 2D and MNIST domains, we trained agents for 5,000 batches; in the Symbolic 4D domain, we trained agents for 20,000 batches. These training times were likely longer than necessary for most of our experiments but 1) established important convergence behaviors and 2) provided a more fair training setting for some of the population baselines, which took longer to converge. In the Symbolic 2D and MNIST domains, there were two training features $|\mathcal{F}| = 2$, while in the Symbolic 4D domain, there were four training features. When generating the data for each batch, we selected the training feature id uniformly at random, and selected input elements uniformly at random (i.e., for symbolic domains, generating the onehot vector for each field randomly, and for the MNIST domains, selecting the digit and fashion images randomly). Each batch therefore contained data for multiple training features.

For our population-based experiments, we used our implementation of the approach presented by Rita et al. (2022). For a population of size $N$, we randomly initialized $N$ speakers and $N$ listeners, each with an associated Adam optimized. With each training batch, we selected a speaker and a listener uniformly at random from the population and trained them using associated optimizers.

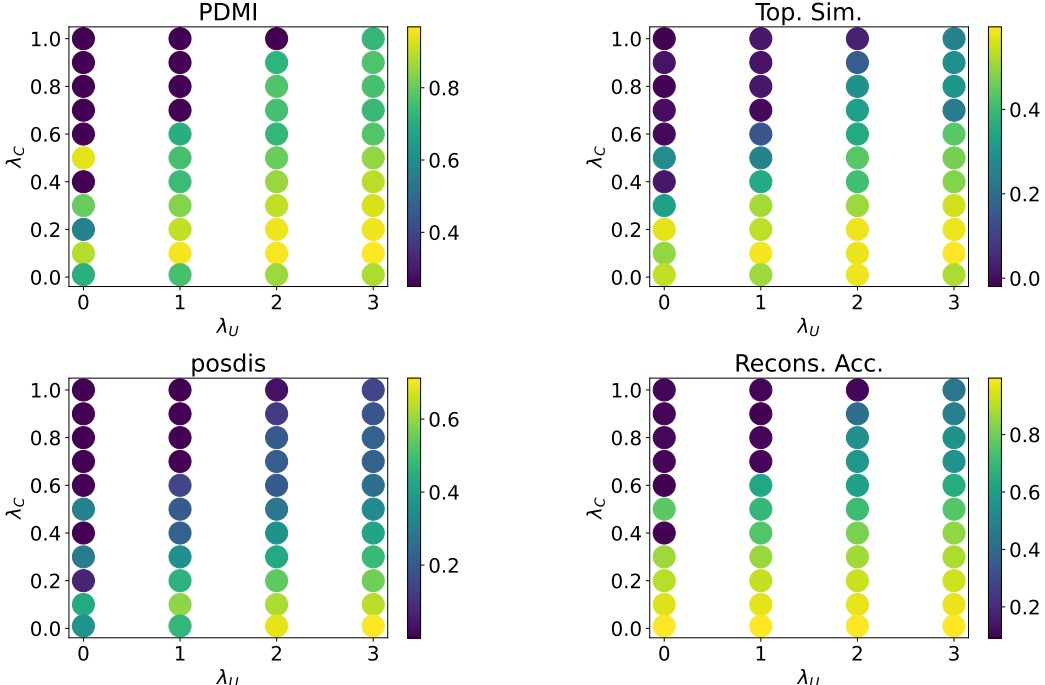

Figure 6: 2D Symbolic VQVIB$_\mathcal{C}$ results for $N = 1, \rho = 1$.

For $\rho! = 1.0$, we randomly selected whether or not to apply gradient updates to the listener with probability $\rho$. We believe this implementation follows from the method outlined in Section 3.4 of Rita et al. (2022), and our convergent results (for $\lambda_C = \lambda_U = 0$) support our belief.

# E    ADDITIONAL RESULTS

In the main paper, we highlighted partial results to illustrate the important role of our complexity-limited multi-task training framework. In particular, we showed results for VQ-VIB$_\mathcal{C}$ models, focusing on `topsim` and `pdmi` metrics. Here, we include further results for various architectures (VQ-VIB$_\mathcal{N}$ and GS) and more metrics (`posdis` and reconstruction accuracy).

## E.1    SYMBOLIC 2D

Compositionality metrics (`pdmi`, `topsim`, and `posdis`), as well as reconstruction accuracy in the Symbolic 2D domain are plotted in Figures 6, 7, 8 for the three speaker architectures we tested. VQ-VIB$_\mathcal{C}$ speakers exhibited the greatest compositionality for small but positive $\lambda_C$. Other speakers had less straightforward trends: compositionality tended to increase with $\lambda_C$ and $\lambda_U$, but at the same time, increasing $\lambda_C$ too much tended to decrease reconstruction accuracy.

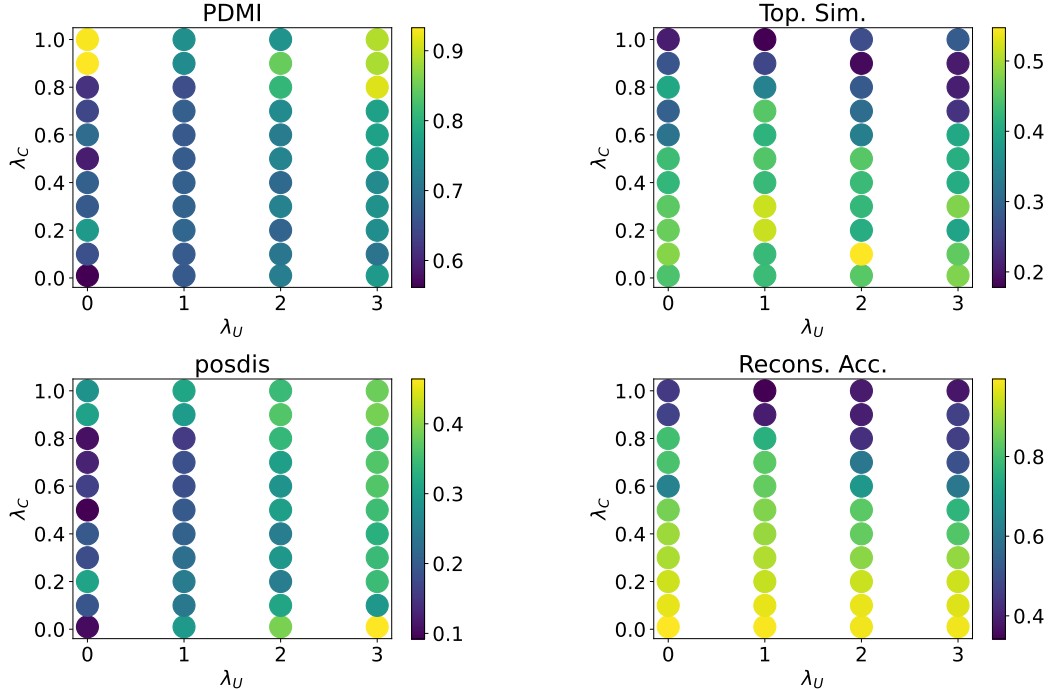

Figure 7: 2D Symbolic VQ-VIB$_\mathcal{N}$ results for $N = 1, \rho = 1$.

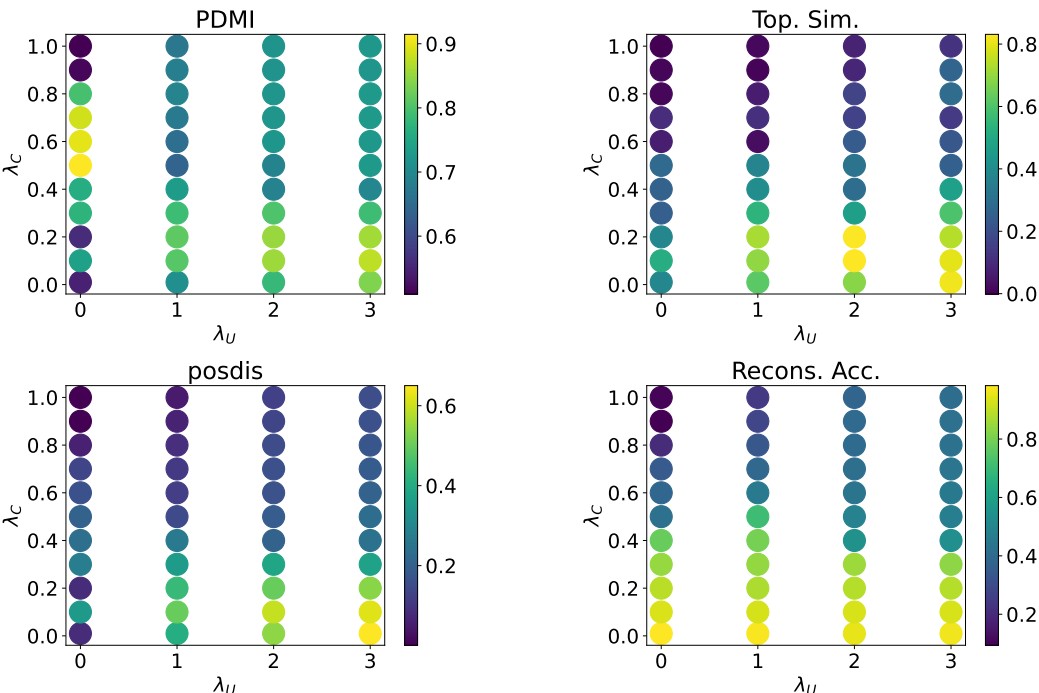

Figure 8: 2D Symbolic GS results for $N = 1, \rho = 1$.

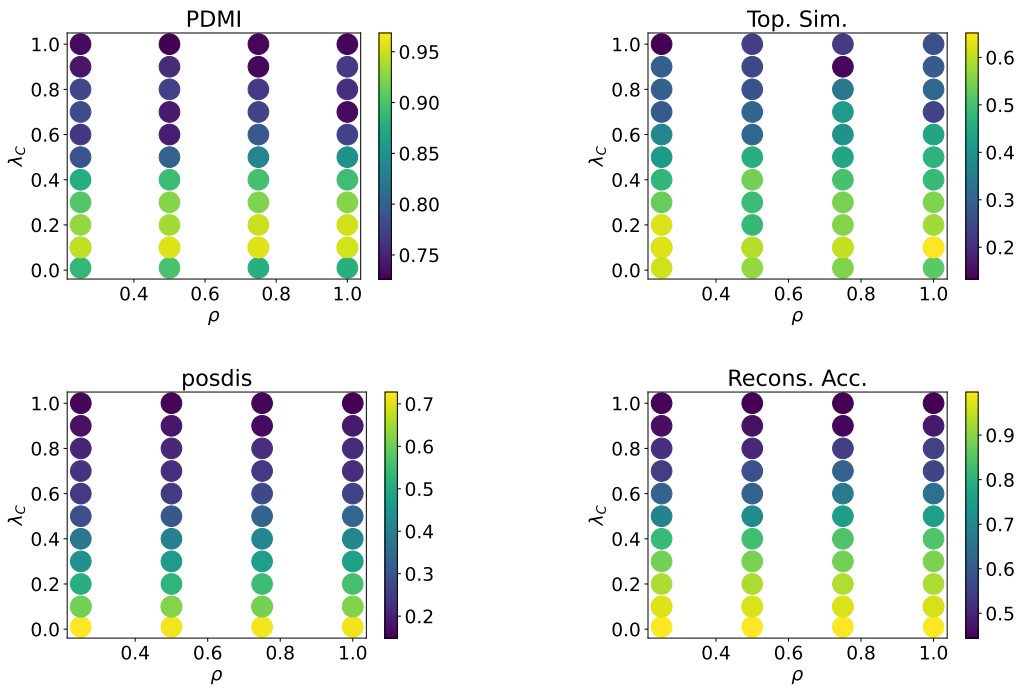

Figure 9: 2D Symbolic VQ-VIB$_\mathcal{C}$ for varying $\rho$. Used $N = 1$ $\lambda_U = 3$.

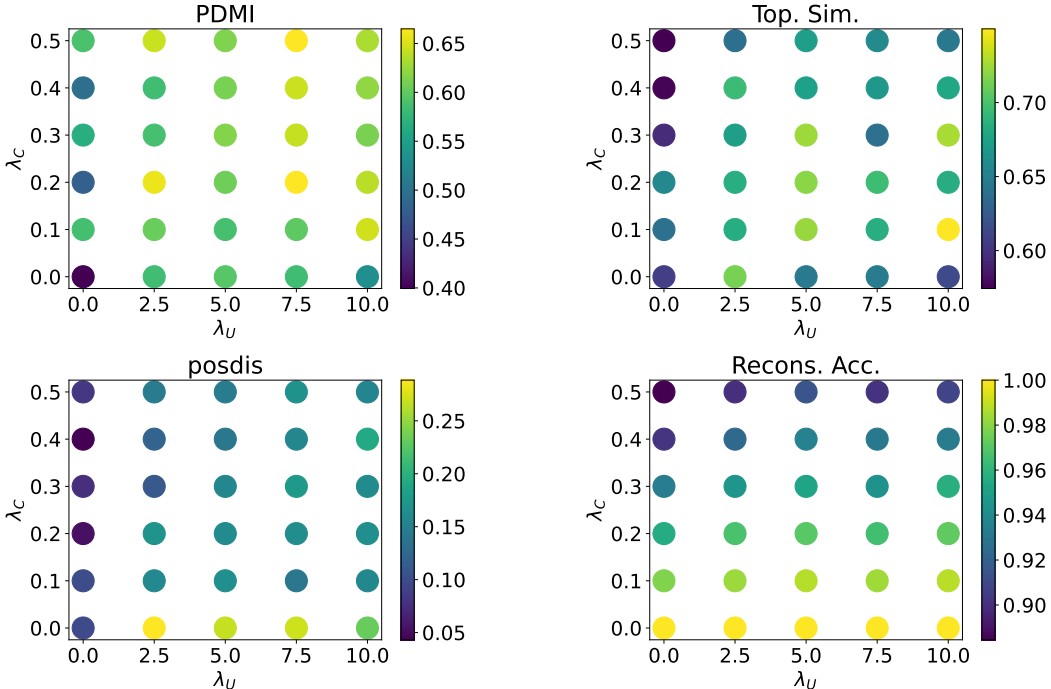

Figure 10: 4D Symbolic VQ-VIB$_\mathcal{C}$ results for $N = 1, \rho = 1$.

## E.2 SYMBOLIC 4D

As in the Symbolic 2D domain, we include complete results for pdmi, topsim, posdis, and reconstruction accuracy for VQ-VIB$_\mathcal{C}$, VQ-VIB$_\mathcal{N}$, and GS speakers in Figures 10, 11, and 12. Interestingly, GS teams did not converge to high reconstruction accuracy when $\lambda_C = 0$. This corroborates some results from prior works that adding some noise in training improves convergence for onehot-based communication (Lowe et al., 2017; Kuciński et al., 2021). We merely highlight that result to emphasize that compositionality metrics for GS agents in that regime should likely be ignored, given the poor overall reconstruction.

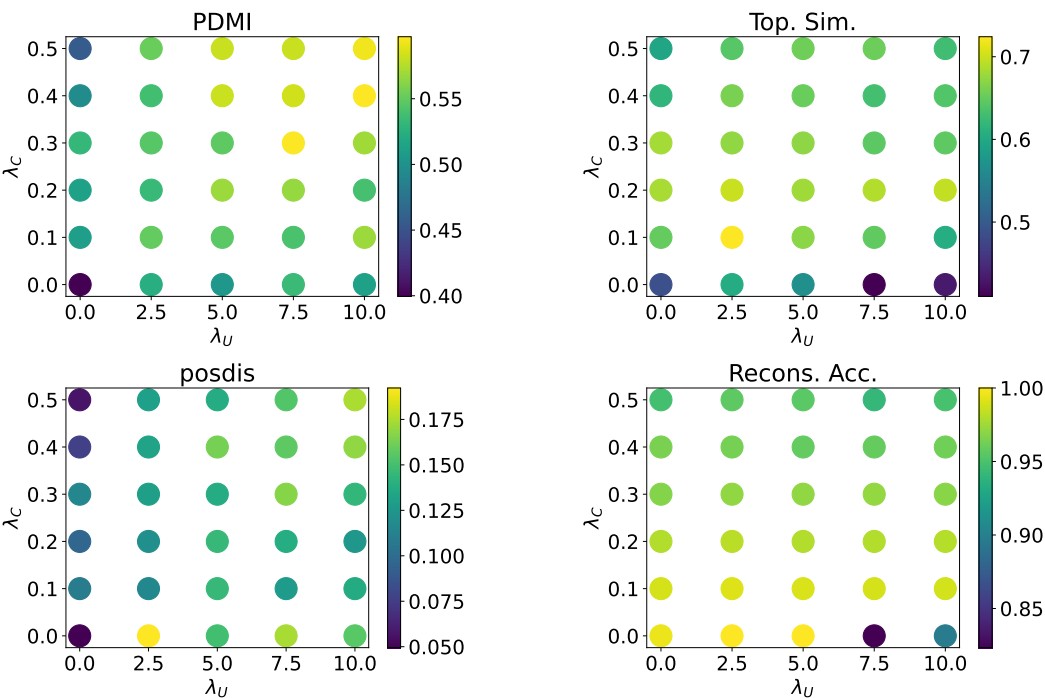

Figure 11: 4D Symbolic VQ-VIB$_{\mathcal{N}}$ results for $N = 1, \rho = 1$.

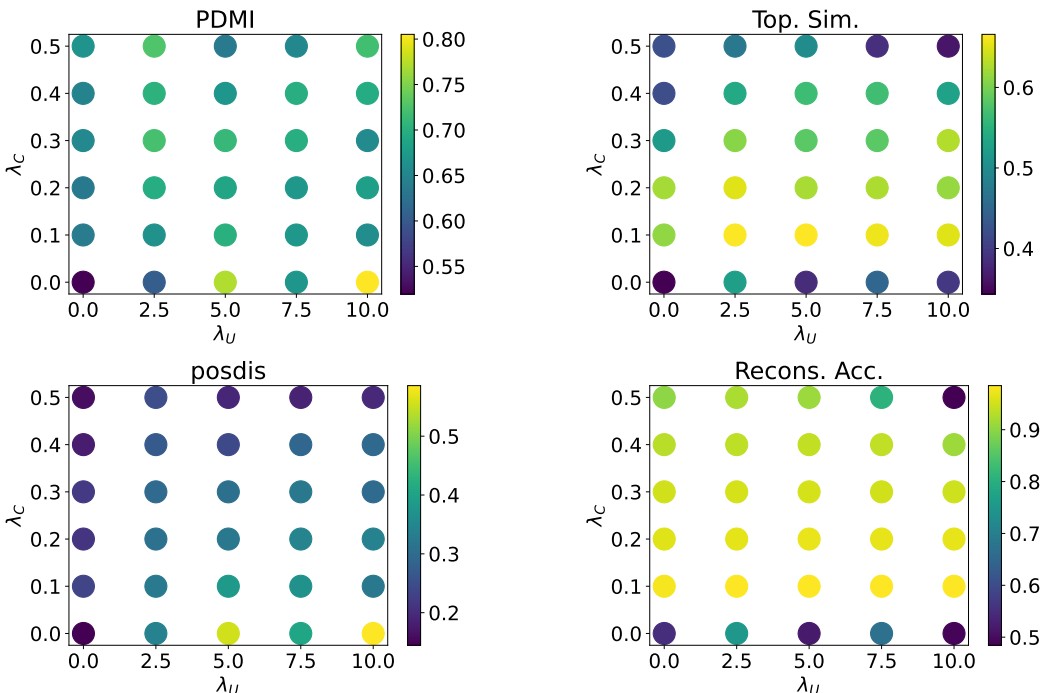

Figure 12: 4D Symbolic GS results for $N = 1, \rho = 1$.

Table 4: Population heterogeneity effects for [4, 4, 4, 4] domain. We include results for $N \in [1, 2, 4, 8]$, complementing the partial results presented in Table 1. All results for $\lambda_U = 10.0$. Medians and standard errors over 10 trials reported. Starred reconstruction accuracy entries indicate that not all 10 trials converged to $> 90\%$ accuracy within the 20,000 training episodes.

| $N$ | $\rho$ | $\lambda_C$ | Recons. | topsim | pdmi | $> 90\%$ Recons. Eps. |
|---|---|---|---|---|---|---|
| 1 | 1 | 0 | 1.00 (0.00) | 0.61 (0.02) | 0.53 (0.01) | 1000 (63) |
| | | 0.1 | 0.99 (0.01) | **0.75 (0.01)** | **0.65 (0.02)** | 1000 (126) |
| | 0.5 | 0 | 1.00 (0.00) | 0.66 (0.02) | 0.57 (0.01) | 1000 (106)) |
| | | 0.1 | 0.99 (0.00) | 0.70 (0.02) | 0.63 (0.02) | 1500 (162) |
| 2 | 1 | 0 | 1.00 (0.00) | 0.65 (0.02) | 0.59 (0.01) | 2500 (242) |
| | | 0.1 | 0.99 (0.00) | 0.74 (0.01) | 0.62 (0.02) | 2500 (174) |
| | 0.5 | 0 | 1.00 (0.00) | 0.66 (0.03) | 0.54 (0.01) | 3000 (318) |
| | | 0.1 | 0.99 (0.00) | 0.72 (0.01) | 0.58 (0.02) | 3000 (126) |
| 4 | 1 | 0 | 1.00 (0.00) | 0.61 (0.02) | 0.54 (0.01) | 7000 (369) |
| | | 0.1 | 0.98 (0.00) | 0.72 (0.01) | 0.58 (0.03) | 6500 (387) |
| | 0.5 | 0 | 1.00 (0.00) | 0.66 (0.02) | 0.56 (0.01) | 6000 (293) |
| | | 0.1 | 0.98 (0.00) | 0.72 (0.01) | 0.62 (0.01) | 8500 (308) |
| 8 | 1 | 0 | 0.99 (0.03) | 0.62 (0.03) | 0.58 (0.01) | 13000* (810) |
| | | 0.1 | 0.96 (0.03) | 0.67 (0.02) | 0.57 (0.04) | 15000* (812) |
| | 0.5 | 0 | 0.97 (0.03) | 0.61 (0.02) | 0.57 (0.02) | 14000 * (668) |
| | | 0.1 | 0.91 (0.04) | 0.65 (0.03) | 0.58 (0.03) | 17000 * 814 |

In Table 1 in the main paper, we presented partial results for population dynamics in the Symbolic 4D domain; here, in Table 4, we include our complete results for more population sizes. Regardless of $N$, we found consistent benefits in increasing $\lambda_C$ from 0.0 to 0.1. This confirms the important role of limiting the complexity of communication. At the same time, overall performance, measured by both reconstruction accuracy and compositionality metrics, tended to worsen as $N$ increased. This appears partially attributable to the slower training associated with larger populations (note the longer convergence times).

### E.3  MNIST

For completeness, we include plots of pdmi, topsim, posdis, and reconstruction accuracy for all three speaker architectures in Figures 13, 14, and 15. Once again, GS teams tended to converge to lower reconstruction accuracies than VQ-VIB models. Interestingly, VQ-VIB$_\mathcal{N}$ agents achieved maximum pdmi for $\lambda_C = 0$ (but still required positive $\lambda_U$). We note that in VQ-VIB$_\mathcal{N}$ method, a clustering loss encourages communication to cluster around discrete tokens; this clustering loss could indirectly encourage less complex communication, which therefore increases compositionality.

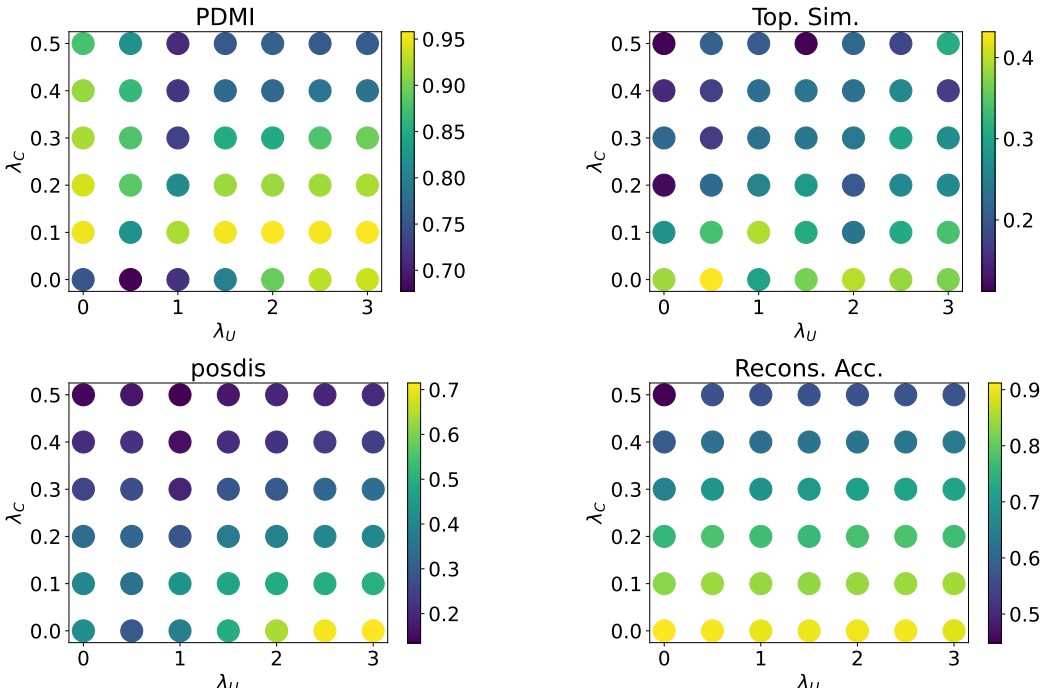

Figure 13: MNIST VQ-VIB$_\mathcal{C}$ results for $N = 1, \rho = 1$.

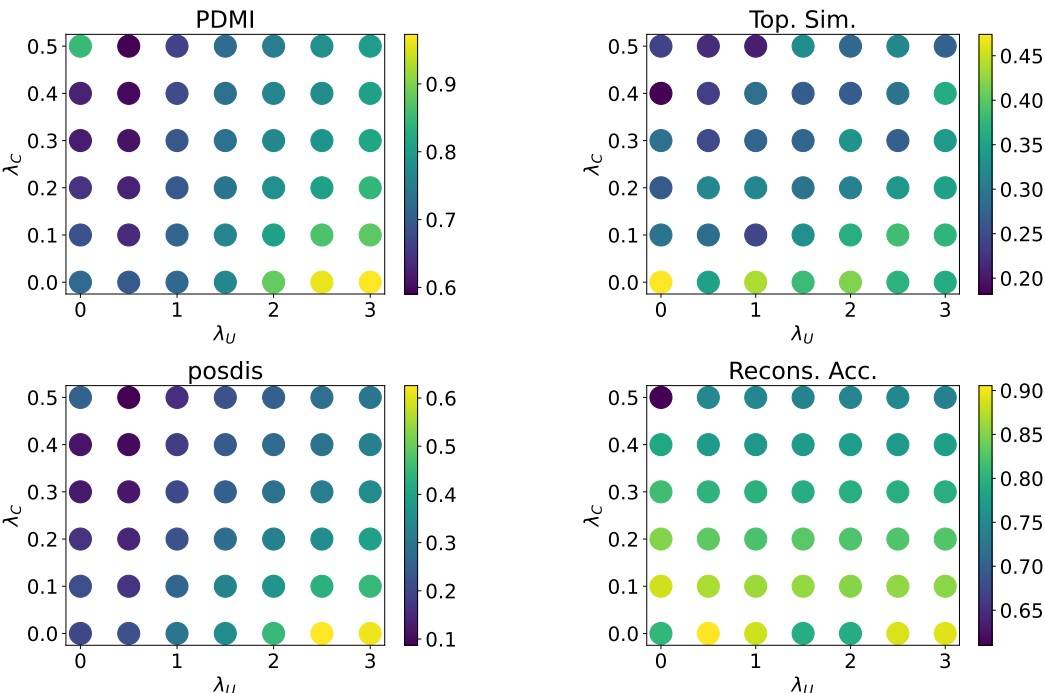

Figure 14: MNIST VQ-VIB$_\mathcal{N}$ results for $N = 1, \rho = 1$.

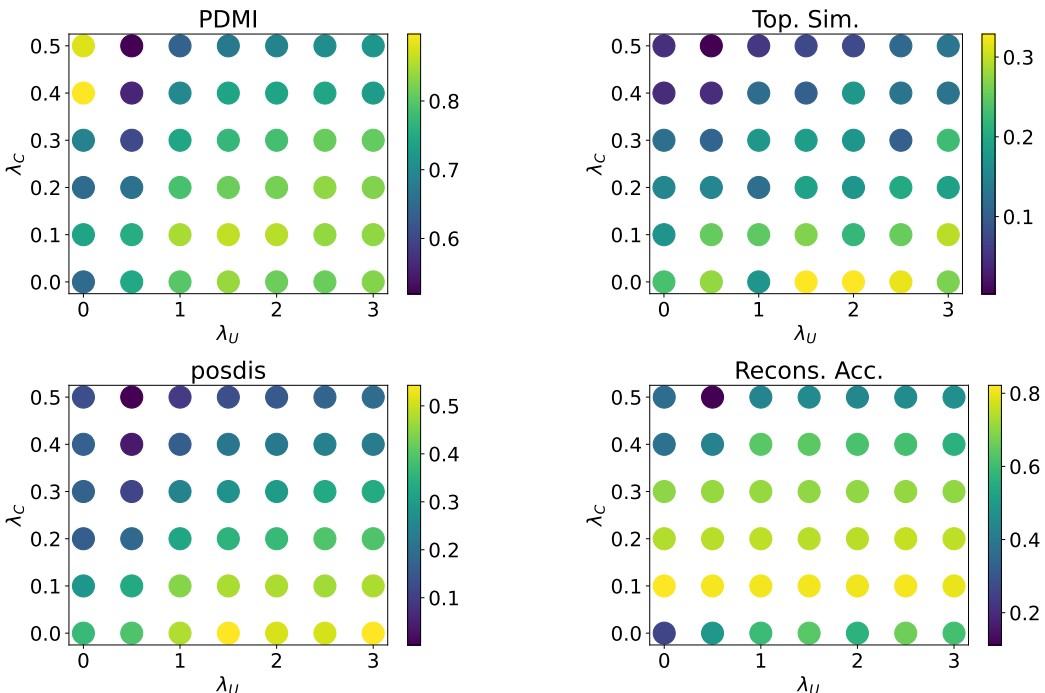

Figure 15: MNIST GS results for $N = 1, \rho = 1$.

Table 5: Mixed Linear Effects Model results for `pdmi` in the Symbolic 2D domain. For $\lambda_C \leq 0.1$, there was a significant increase in `pdmi` when $\lambda_C$ increased (effect size 2.382, $p = 0.002$). Larger $\lambda_C$, however, worsened `pdmi` as reconstructions worsened (effect size -0.74, $p < 0.001$).

|  | Coeff. | Std. Err. | $z$ | $P > |z|$ | [0.025 | 0.975] |
|---|---|---|---|---|---|---|
| Intercept | 0.961 | 0.120 | 8.021 | 0.000 | 0.726 | 1.195 |
| C($\lambda_C \leq 0.1$) | -0.234 | 0.067 | -3.478 | 0.001 | -0.366 | -0.102 |
| $\lambda_C$ | -0.740 | 0.077 | -9.570 | 0.000 | -0.891 | -0.588 |
| C($\lambda_C \leq 0.1$):$\lambda_C$ | 2.382 | 0.773 | 3.083 | 0.002 | 0.867 | 3.896 |

Table 6: Mixed Linear Effects Model results for `pdmi` in the Symbolic 4D domain. The significant ($p = 0.002$) positive interaction $C(\lambda_C \leq 0.1) : \lambda_C$ indicates that `pdmi` increased more for small positive values of $\lambda_C$. Unlike the other two domains, we observed no significant trend for larger $\lambda_C$; we believe that further trials, for larger values of $\lambda_C$, could induce uninformative and non-compositional communication, which would reveal the desired trend.

|  | Coeff. | Std. Err. | $z$ | $P > |z|$ | [0.025 | 0.975] |
|---|---|---|---|---|---|---|
| Intercept | 0.596 | 0.023 | 26.165 | 0.000 | 0.551 | 0.641 |
| C($\lambda_C \leq 0.1$) | -0.064 | 0.020 | -3.223 | 0.001 | -0.102 | -0.025 |
| $\lambda_C$ | 0.035 | 0.046 | 0.772 | 0.440 | -0.055 | 0.125 |
| C($\lambda_C \leq 0.1$):$\lambda_C$ | 0.666 | 0.152 | 4.378 | 0.002 | 0.368 | 0.964 |

## F  STATISTICS TESTS

In the main paper, we presented the partial results of mixed linear effects modeling statistical tests, confirming that small positive $\lambda_C$ induced greater compositionality. Here, for completeness, we include the coefficients and $p-$values for all terms in our fitted statistical models, in all three domains using VQ-VIB$_{\mathcal{C}}$ models, $N = 1$, and $\rho = 1.0$.

In Wilkinson notation, the tests were: `pdmi` $\sim \lambda_C + C(\lambda_C \leq 0.1) + \lambda_C : C(\lambda_C \leq 0.1) + (1|\lambda_U)$ Wilkinson & Rogers (1973). The first three terms model the role of increasing $\lambda_C$ in general, a binary categorical variable representation whether $\lambda_C \leq 0.1$), and the interaction between those two terms. The interaction term shows whether the effect of increasing $\lambda_C$ is different for small and large $\lambda_C$. Lastly, the model groups data by $\lambda_U$ to capture the random-intercept effects of varying the weight on utility.

Results from the Symbolic 2D, Symbolic 4D, and MNIST results are included in Tables 5, 6, and 7, respectively. In all three domains, there is a significant positive interaction ($p \leq 0.002$) between $\lambda_C$ and the categorical variable for $\lambda_C \leq 0.1$. This indicates that `pdmi` increased significantly more in the small $\lambda_C$ region than later, confirming the importance of a small pressure on penalizing complexity.

In both the Symbolic 2D and MNIST domains, we further observed a significant ($p < 0.002$) negative correlation between $\lambda_C$ and `pdmi`, indicating that, for $\lambda_C > 0.1$, increasing $\lambda_C$ worsened compositionality metrics. Such worsening `pdmi` values likely arose due to worsening reconstruction accuracies as $\lambda_C$ grew too large. It is interesting that we did not observe this decrease in `pdmi` in the Symbolic 4D domain, which similarly suffered from decreased reconstruction accuracy, but apparently did not significantly decrease `pdmi`.

Table 7: Mixed Linear Effects Model results for the MNIST domain. As in the Symbolic 2D domain, we observed especially an high interaction effect, indicating that a small increase in $\lambda_C$ led to a large increase in `pdmi`, and a significant negative correlation between `pdmi` and $\lambda_C$ for larger $\lambda_C$.

|  | Coeff. | Std. Err. | $z$ | $P > |z|$ | [0.025 | 0.975] |
|---|---|---|---|---|---|---|
| Intercept | 0.972 | 0.019 | 50.366 | 0.000 | 0.934 | 1.010 |
| C($\lambda_C \leq 0.1$) | -0.144 | 0.016 | -9.174 | 0.000 | -0.175 | -0.114 |
| $\lambda_C$ | -0.419 | 0.035 | -11.814 | 0.000 | -0.489 | -0.350 |
| C($\lambda_C \leq 0.1$):$\lambda_C$ | 1.334 | 0.124 | 10.734 | 0.000 | 1.090 | 1.577 |