# OpenReview forum: "Complexity-Limited Multi-Task Training for Compositional Emergent Communication"
_ICLR.cc/2024/Conference — ICLR 2024 Conference Withdrawn Submission_

### Official Review · Reviewer_mmZM · 2023-10-31

**Soundness:** 2 fair
**Presentation:** 2 fair
**Contribution:** 2 fair
**Rating:** 3
**Confidence:** 3

**Summary:**

This paper discusses the problem of inducing compositional emergent communication in unsupervised neural networks. It proposes a new training method that combines information bottleneck losses with a multi-task framework to encourage compositionality.  Overall, the experiments demonstrated the effectiveness of multi-task training and complexity penalization in inducing compositional communication, with positive effects observed in both symbolic and image domains.

**Strengths:**

1. The paper presents experimental results in different domains, including a symbolic domain and an image domain, which demonstrates the applicability of the proposed framework.
2. The paper mentions the comparison with a baseline method and highlights the advantages of the proposed framework.
3. The paper discusses the trade-off between inducing compositionality and retaining information, which indicates a thoughtful consideration of the research problem.
4. The paper includes statistical tests and analysis to support the findings and conclusions.

**Weaknesses:**

1. The information provided is fragmented and lacks context. It is difficult to understand the specific research question, methodology, and results of the study.
2. The paper does not provide a clear explanation of the symbols and abbreviations used, making it challenging to interpret the information accurately.
3. The paper mentions the use of a multi-task framework and complexity penalization but does not provide detailed explanations or justifications for these approaches.
4. The limitations of the study are briefly mentioned but not thoroughly discussed or analyzed.

**Questions:**

- How do the speaker and listener architectures (Figure 2) contribute to the training process?
- Can you explain the non-monotonic benefits of penalizing complexity and the tradeoff between inducing compositionality and retaining information?
- How do the results from the Symbolic 4D domain and the MNIST image domain support the effectiveness of the proposed approach?
- What is the significance of the theorem on repetition (Theorem 1) in the context of the training framework?

---

### Official Review · Reviewer_yKWc · 2023-10-31

**Soundness:** 2 fair
**Presentation:** 2 fair
**Contribution:** 2 fair
**Rating:** 3
**Confidence:** 1

**Summary:**

This paper proposed a training method called a complexity-limited multi-task training framework. By proposing a metric called Pdmi and comparing it with Posdis, a conventional metric, in three types of tasks, this paper argues that the proposed training method is an effective method for learning compositional emergent communication. .

Various considerations were made by looking at the trends in the value of λ for the three types of tasks. However, although the metric value at each λ is written in the discussion, it was difficult for me to find any quantitative evaluation of the tendency.
Although it is understandable that it is difficult to evaluate using a simple single performance index to express the content that this paper wants to claim, this paper should consider evaluation indicators based on the content of the claims for each experimental result. there were. Therefore, unfortunately, I could not agree that the results of this experiment showed what the author wanted to claim.

**Strengths:**

There were many noteworthy proposals in this paper, including new training methods and metrics.

**Weaknesses:**

The interpretation of the experimental results seemed to lack objectivity. For example, in Figure 3, it is argued that the proposed metric is more sensitive than the conventional metric in terms of the value of λ. Although it certainly gives that impression from the outside, it seems possible to evaluate it using various quantitative indicators, such as calculating the gradient average. I felt that using those numbers in the discussion would make things more objective.

**Questions:**

none

---

### Official Review · Reviewer_Toth · 2023-10-31

**Soundness:** 2 fair
**Presentation:** 2 fair
**Contribution:** 1 poor
**Rating:** 3
**Confidence:** 3

**Summary:**

This paper studies how compositionality can emerge in communication. More specifically, the paper proposes a multi-task setting where a sender must transmit a K-symbol message. An objective is formulated that (1) encourages the reciever to recover various aspects of the original input, (2) encourages the reciever to recover some task-specific aspect of the original input in the smallest possible subset of the K symbols, and (3) maximizes compression of the message using a variational bound. The paper also proposes a new measure of compositionality, pdmi. This measure is used for evaluation along a measure from prior work, topsim. The authors find that the 3 aspects of their proposed objective increase the compositionality of the transmitted messages according to these measures across several toy environments.

**Strengths:**

* This work shows that the proposed objective (which penalizes a measure of complexity) increases various metrics related to compositionality over several toy environments. Understanding the conditions under which compositional communication is encouraged could be relevant to understanding human language or improving generalization of ML systems.

**Weaknesses:**

* My main concern with this paper is the framing of the research questions. It was not clear to me whether the key research questions are well motivated. The primary research question of whether the proposed objective can encourage the encoded messages to be more compositional according to the proposed measure seems a bit circular. It seems somewhat trivial that one can increase various measures of "compositionality" of encoded messages by introducing a regularization objective that closely relates to that desired measure. The information theoretic aspects of the proposed objective seem closely related to aspects of the proposed measure, pdmi. Taken to the extreme, why not regularize directly for pdmi? Other prior work has considered how well communicating agents can generalize to new inputs, and related this to measures of compositionality (e.g. Chaabouni et al. 2020), rather than adopting such measures as the end goal. Does the proposed objective also lead to better generalization? How does pdmi relate to various types of generalization? Alternatively, one could use measures of compositionality to compare and contrast emergent communication between agents to human communication, but the connection between the proposed metric and human-like communication is not well established, and the experiments don't directly offer insights into actual human communication.
* The concept of applying VQ-VIB to emergent communication comes from prior work "Trading off Utility, Informativeness, and Complexity in Emergent Communication" (Tucker et al. 2022). This is (thankfully) made clear in the paper, but does reduce the scope of the contribution.
* Nit: it's not clear to what extent the phenomena described in the paper is "emergent" communication. There is strong supervision for the messages being transmitted.

**Questions:**

* Given the importance of VQ-VIB in this paper, and since it is relatively complex and may be unfamiliar to many readers, it would be helpful to review some of the key concepts in this paper.